# OLIIV Benchmark: Does Your VLM Care What You Say, or How You Say It?

## Abstract

Recent advances in vision-language models (VLMs) have improved their ability to perform multimodal reasoning. However, their capacity to consistently follow answer-specification instructions—explicit directives about how responses should be formatted, structured, or composed—remains largely unexplored. This ability is critical for improving user experience and enabling fair and reliable comparisons across models. To evaluate answer specification instruction following, we introduce OLIIV (Open Language-Image Input Variation), a benchmark designed to measure compliance with answer-specification instructions. OLIIV spans four representative task types: multiple-choice reasoning, binary question answering, structured output generation in JSON, YAML, and XML, and length-constrained image captioning. Each task is tested under systematically varied prompt formulations. This is to assess whether models maintain instruction following compliance when input phrasing changes, but the underlying task remains fixed. Results show that many models perform inconsistently across superficially different, but semantically equivalent, prompts. We found that models often behave differently when presented with Roman numerals versus letters in multiple-choice questions, or produce more compliant Yes/No answers than True/False ones—despite identical instructions. To evaluate adherence to length constraints, we introduce the Length Infidelity Score (LIS)—a deterministic, model-agnostic metric for quantifying over- or under-length responses. Structured-output evaluation further shows that models frequently produce syntactically correct but structurally invalid outputs, such as inserting empty fields or omitting required schema elements. Taken together, our findings reveal that all VLM's used in our experiment are highly sensitive to prompt variation. Such sensitivity limits the fairness of current benchmarking methods, OLIIV fills this gap by providing a structured framework to explicitly test how robust VLMs are to semantically equivalent variations in prompts.

## 1 Introduction

Visual-language models (VLMs) now demonstrate improved multimodal reasoning and perception, allowing them to more effectively interpret and generate content that combines text and images. Although VLMs perform well on tasks like captioning, question answering, and structured data generation, it remains unclear how well they can follow user-specified answer constraints such as specific formats, structures, or length requirements. When models fail to follow user instructions, it can lead to incorrect outputs, require additional manual corrections, and undermine fair comparisons between models because evaluation pipelines adapt the prompt per model to ensure compliance.

Existing benchmarks primarily evaluate task correctness [Mizrahi et al. (2023); Zhou et al. (2023); Qin et al. (2024)], but often fail to assess whether models follow explicit response constraints—such as required formats, structural accuracy, or length limits. In practice, users frequently work around these shortcomings through prompt engineering or custom parsing scripts. However, this masks issues in instruction-following and obscures which models reduce the burden on users. Without benchmarks that directly measure instruction adherence, it is difficult to determine whether models are genuinely robust to prompt variations or simply memorizing familiar response patterns. This calls for a systematic means of measuring of prompt sensitivity for VLMs specifically.

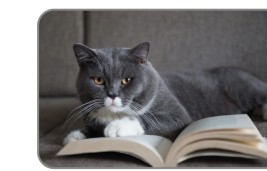

**Category: Binary**

**Prompt A:**
Does the image show a cat?
Please answer Yes or No only.
**Expected Answers:** "Yes" or "No"

**Prompt B:**
Please answer True or False according to the following statement about the image.
The image shows a cat.
**Expected Answers:** "True" or "False".

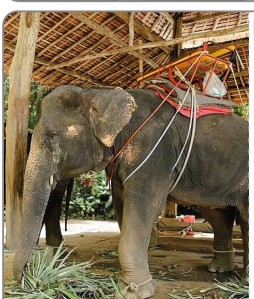

**Category: Multiple Choice**

**Prompt A:**
Select only the letter inside the parentheses corresponding to the most appropriate image caption for the given image from the following options:
(A) A man leans forward and hits a ping pong ball.
(B) Elephant with chair strapped to back and mouth open.
(C) The batter swings his bat and hits the ball.
(D) Two men with their heads touching, very intimate.
**Expected Answers:** "A" or "B" or "C" or "D".

**Prompt B:**
Select only the Roman number inside the parentheses corresponding to the most appropriate image caption for the given image from the following options:
(I) A man leans forward and hits a ping pong ball.
(II) Elephant with chair strapped to back and mouth open.
(III) The batter swings his bat and hits the ball.
(IV) Two men with their heads touching, very intimate.
**Expected Answers:** I" or "II" or "III" or "IV".

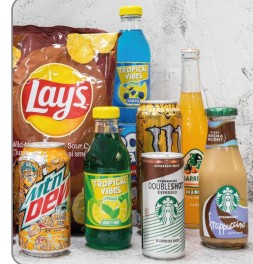

**Category: Structured**

**Prompt A:**
Write an image caption with no more than 1 sentence and a list of ≥ 2 objects present in the image. Your output must be a parseable JSON string that the python function json.loads() can load successfully. Also, your output JSON string must contain only the following keys: "image_caption", "objects". ["object 1", "object 2", "object 3", "object 4"]
**Expected Answers:** Parseable JSON with requested fields and formats

**Prompt B:**
Write an image caption with no more than 1 sentence and a list of ≥ 2 objects present in the image. Your output must be a valid YAML format that the python function http://yaml.safe_load() can successfully load the content. Your output should contain only the following YAML structure: caption: \tThe image caption \nobjects:\n \t-object 1\n\t-object 2\n\t-object 3\n\t-object 4
**Expected Answers:** Parseable YAML with requested fields and formats

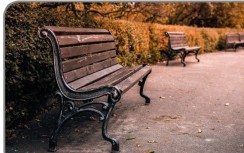

**Category: Textual Constrained**

**Prompt A:**
Write an image caption using no more than 11 words.
**Expected Answers:** Caption with no more than 11 words.

**Prompt B:**
Write an image caption using no more than 20 words.
**Expected Answers:** Caption with no more than 20 words.

Figure 1: Visual-language models (VLMs) often struggle to follow specific response requirements, such as providing answers in a particular format. This forces users to rely on prompt engineering or manual post-processing. It reduces usability and makes fair model comparison difficult. To address this, we introduce OLIIV—a benchmark designed to evaluate instruction-following compliance across four task categories: (1) Binary questions (e.g., Yes/No), (2) Multiple-choice formats, (3) Structured outputs (e.g., JSON, YAML), and (4) Text length constraints. Each task includes controlled variations in prompt phrasing and formatting to test whether models can generalize beyond superficial differences. The figure illustrates examples of these instruction types for VLMs.

To address these limitations, we introduce OLIIV, a benchmark designed to evaluate instruction-following behavior across four task types: (1) multiple-choice reasoning with lettered and Roman numeral formats; (2) binary question answering using Yes/No and True/False variants; (3) structured output generation in JSON, YAML, and XML; and (4) length-constrained image captioning. We selected these tasks because they reflect common, real-world scenarios where users expect models to follow specific formatting, structure, or length constraints (most multi-modal knowledge benchmarking use multiple-choice [He et al. (2024); Lu et al. (2022); Wang et al. (2024a); Li et al. (2023a); Yue et al. (2024a); Lu et al. (2023)], caption lengths vary for accessibility or descriptive applications [Dwibedi et al. (2024); Kastner et al. (2021); Ding et al. (2023); Hirsch & Tal (2022)], and structured format is useful for pseudo-labeling) . We evaluate each task using systematic prompt variations to test how changes in formatting affect a model's ability to follow instructions, even when the underlying task and expected answer stay the same (see Figure 1).

A core contribution of OLIIV is its focus on testing whether VLMs can follow instructions consistently when prompt wording or formatting changes (see Figure1). This is because a systematic measure of prompt sensitivity, for VLMs specifically, is currently underexplored by current research

in the benchmarking space. We found that many models gave longer answers when multiple-choice options used Roman numerals instead of letters, even though the instructions were identical. Surprisingly, models followed instructions more accurately in open-ended True/False questions than in multiple-choice ones that explicitly required concise answers. These results suggest that models often rely on familiar formatting patterns rather than truly understanding and following user instructions.

We also found that models often ignored formatting instructions in structured output tasks, defaulting to familiar response patterns. GPT-4o [OpenAI (2024)], inserted empty fields in 74% of JSON responses. Phi-3.5, Phi-4.0 [Gunasekar et al. (2024)], and InternVL2.5-8B [Chen et al. (2023)], failed YAML generation due to indentation and hierarchy errors. LLaMA-3.2-11B [AI (2024)], often added explanatory text or invalid punctuation, and omitted critical XML headers in 24% of cases. Similarly, Intern-3-1B [Chen et al. (2023)] and Phi-3.5 [Gunasekar et al. (2024)] omitted XML declarations in 97% and 71% of cases, respectively. These errors indicate format-specific overfitting, and call into question the reliability of structured output generation in VLMs.

To assess instruction-following in Multiple Choice, Binary QA, and Image Captioning tasks, we introduce the Length Infidelity Score (LIS)—a deterministic, model-agnostic metric that measures how much a model's response deviates from the expected length. These tasks often require precise output formats—such as a single letter, a one-word answer, or a caption within a set length—where even minor deviations can reduce usability and complicate fair model comparison. LIS provides a consistent and interpretable way to evaluate compliance with these constraints, even when prompt wording varies. In cases where these tasks also involve structured outputs, we extend our evaluation by identifying specific types of failures—such as missing fields, formatting errors, or invalid values. This detailed analysis captures instruction-following breakdowns that binary correctness metrics like IfEval[ Zhou et al. (2023)] overlook, offering a deeper understanding of model behavior.

Our evaluations across proprietary models and open-source models showed that compliance failures are not solely due to task complexity, but are influenced by how prompts are phrased and formatted. These findings highlight the importance of assessing models not just on task accuracy, but on their ability to follow instructions consistently across variations. OLIIV addresses this gap by explicitly measuring robustness to prompt variation and fidelity in structured responses—offering a more realistic and diagnostic benchmark for instruction-following performance in VLMs.

## 2 RELATED WORK

Benchmarks measuring the performance of VLMs is crucial to shape the development of the field, as stated by Patterson Patterson (2012). In the realm of multimodal models, there is a vast amount of benchmarks dedicated to measure performance across scientific topics, visual perception tasks such as chart and/or diagram understanding, among others - whereas OLIIV focuses on compliance under systematic prompt variation. This section reviews general benchmarks as well as the instruction-following benchmarks on visual-language tasks.

**General Broad-Topic Benchmarks:** These benchmarks have focused mainly on measuring the performance across scientific topics such as Mathematics, Chemistry, Biology among other fields Yue et al. (2024a;b); Lu et al. (2023); Li et al. (2024b); Singh et al. (2019); Li et al. (2024b); diagrams and chart understanding Kembhavi et al. (2016); Wang et al. (2024b); Masry et al. (2022); and visual perception tasks Liu et al. (2024b); Kembhavi et al. (2016); Yu et al. (2023); Goyal et al. (2017); Padlewski et al. (2024); Li et al. (2024a); Liu et al. (2023b) such as image captioning Lin et al. (2014), video generation Huang et al. (2024), object descriptions Li et al. (2023c); Mao et al. (2023), OCR Yang et al. (2024); He et al. (2018), and visual-temporal perception tasks Damen et al. (2018); Grauman et al. (2022); Li et al. (2024c; 2021); and reasoning Han et al. (2023).

**Multimodal Instruction-Following Benchmarks:** Unlike existing instruction-following benchmarks Xu et al. (2022); Qian et al. (2024); Bitton et al. (2023); Epstein et al. (2024); Dai et al. (2023); Li et al. (2023b); Xu et al. (2024); Liu et al. (2023a) that mainly focus on image, video question-and-answer instructions, OLIIV focuses on measuring the compliance of the model with the instructions describing expected properties of the answer (, structured answers, formats, length). MIABench Qian et al. (2024) tests for image descriptions of a portion of an image, length limit on answer, mentioning of objects in answer, answer styles, grammar, math, perspective as in opinion,

and OCR tasks. VisIt Bitton et al. (2023) tests for instruction following on mainly image captioning. The benchmark includes challenging prompts to describe images including instruction-conditions captions. MMMT Epstein et al. (2024) benchmarks multi-turn instruction following benchmark. Unlike MIABench, VisIT, and MMT, OLIIV measures stricter compliance with the instructions given to the VLMs on how to answer. To visualize the differences, see Figure 3 in the Appendix.

## 3 BENCHMARK

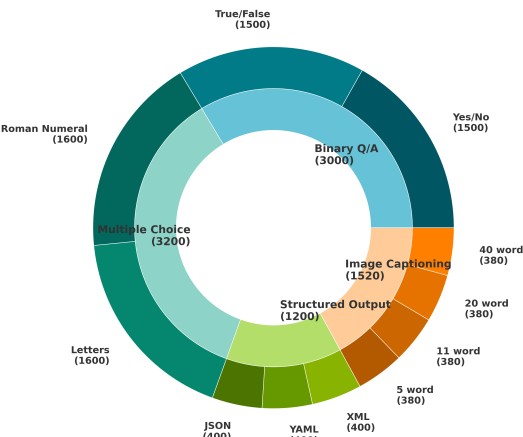

Figure 2: Question-Type Breakdown: OLIIV contains 8,920 total prompts, across categories of tasks: Multiple Choice, Binary Q/A, Image Captioning, and Structured Output. Each of these tasks contains sub-categories which constitute the input variations within in each task type. Multiple Choice tasks uses both Roman Numerals and Capital letters. Binary Q/A tasks use both True/False and Yes/No variations. Structured Output tasks pasn JSON, YAML and XML. Image Captioning uses different maximum word counts: 5, 11, 20 and 40.

The primary goal of OLIIV is to assess how VLMs follow answer-specification instructions in the prompts for visual-language tasks. This is to address a notable gap in the current benchmarking space for a systematic measure of prompt sensitivity: for VLMs specifically. To this end, we collected 746 images from mainly two datasets: 1) COCO Lin et al. (2014); and 2) No-Caps Agrawal et al. (2019). We generated a diverse set of prompts to evaluate instruction-following capabilities across a range of response formats. The benchmark includes **8,920 evaluation prompts**, organized into four task categories: *Binary Q/A (BQA)*, *Multiple Choice (MCQA)*, *Structured Output*, and *Image Captioning*. **Binary Q/A**: 3,000 prompts, evenly split between *Yes/No* and *True/False* conditions (1,500 each). **Multiple Choice**: 3,200 prompts, with two formatting conditions: *Roman Numerals* and *Capital Letters* (1,600 each). **Structured Output**: 1,200 prompts spanning three formats—*JSON*, *YAML*, and *XML*—with 400 prompts per format. **Image Captioning**: 1,520 prompts across four word-length conditions, with 380 prompts per condition. See Figure 2 for a visual breakdown.

OLIIV evaluates four common VLM tasks: (1) image captioning, (2) binary questions (requiring "Yes"/"No" or "True"/"False" responses), (3) multiple-choice questions selecting from predefined options, and (4) structured outputs - adhering to JSON, YAML, or XML formats (see Figure 2). To ensure robustness, each task in OLIIV was chosen to focus on a different aspect of instruction compliance, allowing the identification of systematic model failures.

For each task, we generate multiple prompt variations to test whether models follow answer specifications under semantically equivalent variations (examples in Figure 1, full set in Appendix). Prompts query visual information from dataset annotations (COCO Lin et al. (2014), No-Caps Agrawal et al. (2019)). For captioning, we enforce word limits of 5, 11, 20, and 40; for structured output, we test adherence to JSON, YAML, and XML schema constraints. BQA is evaluated across "Yes/No" vs. "True/False," and MCQA across lettered vs. Roman numeral options. OLIIV's BQA and MCQA suites address the lack of uniformity in VLM evaluation, where models often ignore answer instructions. Existing benchmarks (MMMU, MME, MathVista, etc.) [Yue et al. (2024a); Lu et al. (2023); Fu et al. (2024)] instead rely on parsing code and prompt adaptations—for instance, MMMU and MMMU Pro provide MCQA parsing scripts [Yue et al. (2024a;b)], while LMMS-eval offers parsing functions and model-specific adaptations (e.g., AI2D prompts, MathVista prompts for Qwen-VL [Bai

et al. (2023); Kembhavi et al. (2016)]). OLIIV further includes five additional suites: expanded prompt variations for MCQA and BQA (forgreater linguistic diversity and paraphrasing), plus strict-instruction variants of MCQA, BQA, and captioning tasks to test sensitivity to strongly emphasized constraints.

To quantify instruction-following compliance in the Image Captioning, Binary Questions, and Multiple Choice tasks, OLIIV introduces the **Length Infidelity Score (LIS)** (see Section 3.1). In the evaluation of structured output tasks we characterize response compliance by evaluating distinct error types, including missing fields, formatting errors, invalid values, and parsing failures. Unlike IfEval Zhou et al. (2023), which reduces performance to a binary pass/fail score, our framework provides a structured, interpretable breakdown of failure modes, enabling more precise comparisons across models and error categories.

### 3.1 LENGTH-INFIDELITY-SCORE (LIS)

Length is a direct and easily quantifiable signal of instruction-following behavior. In tasks like image captioning, where models are instructed to produce outputs within a strict word limit, any deviation in length immediately reflects a failure to comply with the specified constraint, regardless of semantic accuracy. Similarly, in fixed-option tasks such as multiple-choice or binary Q/A, instructions often explicitly request a single word or letter (e.g., "A" or "Yes"). Verbose or multi-sentence responses violate these expectations, even if the underlying answer is correct. Because length deviations are unambiguous and interpretable, they offer a reliable proxy for measuring whether models are obeying structural aspects of the prompt. To evaluate compliance in such cases, we introduce the **Length Infidelity Score (LIS)**, a metric that quantifies deviations from expected response lengths. It is fully deterministic, model-agnostic, and operates solely on model outputs—requiring no introspection, calibration, or learned scoring components—making it both reproducible and easy to compute across evaluation settings. LIS is used in OLIIV to measures whether models respect length-based constraints as a proxy for instruction adherence. The LIS metric is defined as follows:

$$\text{LIS} = \frac{1}{n \cdot l_{\text{upper}}} \sum_{i=1}^{n} \max(l_i - l_{\text{upper}}, 0) \tag{1}$$

where $n$ refers to the number of records, $l_i$ refers to the length of response i, and both $l_{\text{lower}}$ and $l_{\text{upper}}$ are the lower and upper bounds for the response length, respectively. Intuitively, it computes the average amount of words or characters the responses of a model are out of bounds by, with respect to the upper-bound. Note that a perfect score is thus **0**, and that **the higher the score, the less the model is in compliance** with length requirements.

## 4 EXPERIMENTS

We evaluated performance on OLIIV using a suite of both commercial and open-source models: GPT-4o, GPT-4o Mini, Gemini-2.0 Flash Lite, Gemini-2.0 Flash, Gemini-1.5 Pro, Gemini-1.5 Flash, Claude-3-7 Sonnet, Claude-3.5 Sonnet, Intern-3-8B, Intern-3-1B, Intern-2.5-7B, Intern-2.5-1B, Llama-3.2-11B-VI (Vision Instruct), Llava-Next, Phi-4, Phi-3.5, Qwen2-VL-7B (Instruct), QwenVL, [OpenAI (2024); AI (2025); DeepMind (2024); Anthropic (2024); Chen et al. (2023); AI (2024); Liu et al. (2024a); Gunasekar et al. (2024); Inc. (2023)]. All models were queried with a temperature of 0.0 to ensure deterministic outputs and support reproducibility. Commercial models were accessed via their official APIs—Anthropic, Google Generative AI, and OpenAI—while open-source models were executed using the Hugging Face Transformers library. For each model, we collected Length Infidelity Scores (LIS) (see Eq. 1) for their responses. LIS measures the deviation of a model's response from user-specified length constraints, and was evaluated across the multiple-choice, binary question-answer, and image captioning prompt sets (see Table 1). For these test suites, we also ran the expanded set of prompts to evaluate how strictness and greater prompt diversity affect responses. We also analyzed structured output responses by identifying specific failure modes (, field omissions, formatting errors) for each model. In addition to these, we report baseline performance metrics such as accuracy and BERTScore for broader context (see Supplementary Materials: measuring task-performance is not the goal of this research, and is thus not included here). Finally, as OLIIV is a relatively lightweight benchmark, all experiments were performed on local CPU's, and the experiments have minimal computational requirements.

Table 1: Length Infidelity Scores (LIS) for multiple-choice and binary-answer tasks across different prompt formats ($l_{upper} = 1$). Higher LIS values indicate greater verbosity: i.e., the model produced additional text beyond the instructed response. Thus, **the higher the LIS value, the less the model is in compliance**. Perfect scores of 0.0 are green and become increasingly red-shifted as compliance decreases.

| Model | Multiple-Choice | | Bin-Answer | | | MC (Strict) | | Bin (Strict) | |
|---|---|---|---|---|---|---|---|---|---|
| | Letter | Roman | Y/N | T/F | TF-NoBound | Letter | Roman | Y/N | T/F |
| GPT-4o | 2.17 | 4.13 | 0.00 | 0.00 | 3.17 | 0.01 | 0.00 | 0.00 | 0.00 |
| GPT-4o Mini | 0.70 | 3.96 | 0.00 | 0.00 | 0.11 | 0.00 | 0.00 | 0.00 | 0.00 |
| Gemini-2.0 Flash Lite | 11.64 | 5.11 | 0.00 | 0.10 | 12.95 | 0.00 | 0.00 | 0.00 | 0.00 |
| Gemini-2.0 Flash | 11.65 | 5.15 | 0.00 | 0.07 | 12.93 | 0.01 | 0.00 | 0.00 | 0.00 |
| Gemini-1.5 Pro | 11.67 | 5.12 | 0.00 | 0.13 | 13.60 | 0.00 | 0.00 | 0.00 | 0.00 |
| Gemini-1.5 Flash | 11.73 | 5.13 | 0.00 | 0.12 | 12.92 | 0.00 | 0.00 | 0.00 | 0.00 |
| Claude-3-7 Sonnet | 11.69 | 13.33 | 1.82 | 14.33 | 15.48 | 0.04 | 0.03 | 0.00 | 0.00 |
| Claude-3.5 Sonnet | 25.34 | 26.76 | 1.34 | 12.91 | 15.17 | 0.00 | 0.00 | 0.00 | 0.00 |
| Intern-3-8B | 6.38 | 14.36 | 0.00 | 0.00 | 0.00 | 0.24 | 0.01 | 0.00 | 0.00 |
| Intern-3-1B | 5.75 | 5.33 | 0.00 | 0.00 | 0.00 | 3.72 | 2.28 | 0.00 | 0.00 |
| Intern-2.5-7B | 6.68 | 4.77 | 0.00 | 0.00 | 0.08 | 2.19 | 0.01 | 0.00 | 0.00 |
| Intern-2.5-1B | 6.89 | 7.16 | 0.00 | 0.00 | 0.00 | 4.86 | 1.52 | 0.00 | 0.00 |
| Llama-3.2-11B-VI | 8.70 | 7.35 | 0.00 | 0.00 | 23.83 | 0.03 | 0.03 | 0.00 | 0.00 |
| Llava-Next | 0.20 | 2.98 | 0.00 | 0.00 | 0.00 | 0.03 | 0.11 | 0.00 | 0.00 |
| Phi-4 | 0.60 | 2.12 | 0.00 | 0.00 | 0.00 | 0.00 | 0.00 | 0.00 | 0.00 |
| Phi-3.5 | 0.00 | 0.38 | 0.00 | 0.00 | 0.00 | 0.00 | 0.03 | 0.00 | 0.00 |
| Qwen2-VL-7B | 3.13 | 5.89 | 0.00 | 0.00 | 0.00 | — | — | — | — |
| QwenVL | 3.13 | 5.89 | 0.00 | 0.00 | 0.00 | 0.13 | 0.48 | 0.00 | 0.00 |

## 4.1 MULTIPLE CHOICE

In our Multiple Choice prompt bed we use four prompt variations for each image (see supplementary material for all variations). Each of which asks the model to identify, from a selection of options, the correct answer of which object is in the image. Two of these variations use capital letters, and the other two use Roman numerals. *Importantly, every prompt asks the model to respond by **only** selecting the correct letter/roman numeral.* A model's LIS value in this case captures how much additional information it is providing, and thus quantitatively measures the extent to which it is non-compliant with the prompt's explicit requirement. LIS calculates this on the wordcount level, in the case of Multiple Choice selections, only **one** word should be present. Thus, we set the upper-bound to one.

Comparing these two prompt-types reveals that almost of all the run models are sensitive to the superficial change of Roman Numerals and Capital Letters (see Table 1). Openai's GPT-4o and GPT-4o Mini, as well as InternVL are more verbose (and thus less compliant) with Roman numerals than capital letters, while the opposite is true of Google's Gemini models and Phi-4.0. Claude-3.5 Sonnet is the least compliant with LIS values that more than double the other models, followed by Claude-3.7 Sonnet. Only Phi-3.5 performs universally well. Finally, we can also see that Multiple-Choice compliance is highly sensitive to strictness in almost all models.

## 4.2 BINARY ANSWERS

In the Binary-Answer prompt bed, we use two variations per image: one prompting a Yes/No response, and the other prompting True/False (see Figure 1). As with the Multiple-Choice prompt bed, compliance is measured using Length Infidelity Score (LIS), computed at the word-count level. We set the upper-bound to one. To isolate the effect of instruction specificity, we also include a third experimental condition that mirrors the True/False prompt but omits any explicit instruction to limit the response to either "True" or "False." This condition serves as a comparative baseline for evaluating whether compliance is driven more by explicit instructions or by superficial variations of the prompt. Unlike with the Multiple Choice condition, for most models there is only *slight* bias with Yes/No responses generally containing fewer extra words than True/False responses (see Table 1).

Overall, Binary-Answer responses are more concise and compliant than Multiple-Choice ones. This holds under strictness, and expanded prompt variations show the same trend (see Appendix). However, a closer comparison between Binary-Answer and Multiple-Choice LIS scores reveals unexpected trends. GPT-4o, GPT-4o Mini, Claude-3.5 Sonnet, all Intern models, LLaVA-Next, and all Qwen and Phi variants outperform in the True/False-NoLimit condition than in either Multiple-Choice format. This is counterintuitive: Multiple-Choice prompts explicitly require a single option, yet models

Table 2: LIS scores for word-count-constrained image caption task across different target lengths. We tested the following maximum wordcounts: 5, 11, 20, 40. Compliance improves as the maximum wordcount increases, with most models struggling in strictest condition. $l_{upper}$ is 5, 11, 20, 40, respectively.

| Model | 5-word | 11-word | 20-word | 40-word | Strict–5w | Strict–11w | Strict–20w | Strict–40w |
|---|---|---|---|---|---|---|---|---|
| GPT-4o | 1.02 | 0.06 | 0.00 | 0.00 | 0.00 | 0.00 | 0.00 | 0.00 |
| GPT-4o Mini | 0.97 | 0.06 | 0.00 | 0.00 | 0.00 | 0.00 | 0.00 | 0.00 |
| Gemini-2.0 Flash Lite | 0.29 | 0.14 | 0.00 | 0.00 | 0.01 | 0.00 | 0.00 | 0.00 |
| Gemini-2.0 Flash | 0.24 | 0.14 | 0.00 | 0.00 | 0.00 | 0.00 | 0.00 | 0.00 |
| Gemini-1.5 Pro | 0.32 | 0.08 | 0.00 | 0.00 | 0.01 | 0.00 | 0.00 | 0.00 |
| Gemini-1.5 Flash | 0.25 | 0.06 | 0.00 | 0.00 | 0.00 | 0.00 | 0.00 | 0.00 |
| Claude-3-7 Sonnet | 1.60 | 0.07 | 0.00 | 0.00 | | | | |
| Claude-3.5 Sonnet | 1.40 | 0.05 | 0.00 | 0.00 | | | | |
| Intern-3-8B | 1.39 | 0.10 | 0.00 | 0.00 | 1.16 | 0.00 | 0.00 | 0.00 |
| Intern-3-1B | 3.00 | 0.50 | 0.00 | 0.00 | 0.29 | 0.08 | 0.20 | 0.04 |
| Intern-2.5-7B | 1.39 | 0.06 | 0.00 | 0.00 | 0.04 | 0.01 | 0.00 | 0.00 |
| Intern-2.5-1B | 1.00 | 0.10 | 0.00 | 0.00 | 0.11 | 0.05 | 0.01 | 0.00 |
| Llama-3.2-11B-VI | 1.60 | 0.25 | 0.00 | 0.00 | 0.12 | 0.02 | 0.00 | 0.01 |
| Llava-Next | 0.59 | 0.16 | 0.00 | 0.00 | 0.16 | 0.00 | 0.00 | 0.00 |
| Phi-4 | 1.00 | 0.04 | 0.00 | 0.00 | 0.21 | 0.00 | 0.00 | 0.00 |
| Phi-3.5 | 0.59 | 0.11 | 0.00 | 0.00 | 0.19 | 0.07 | 0.03 | 0.00 |
| Qwen2-VL-7B | 1.80 | 0.13 | 0.00 | 0.00 | | | | |
| QwenVL | 1.80 | 0.13 | 0.00 | 0.00 | 0.67 | 0.03 | 0.00 | 0.00 |

produce longer, more verbose outputs than for True/False-NoLimit, which includes no such directive. Several Gemini models also generate Letter responses nearly as verbose as their True/False-NoLimit outputs. This suggests that formatting expectations influence compliance more than explicit rules: if explicit instructions sufficed, Multiple-Choice outputs should match Binary-Answer brevity; instead, models align more with binary structural conventions than with stated task rules. This not only indicates that adherence depends on both prompt content and learned biases tied to answer type, but quantitatively captures the extent to which this manifests across models: highlighting how OLIIV captures sensitivity to shallow linguistic variation and instruction-following failures.

### 4.3 WORD-COUNT-CONSTRAINED IMAGE CAPTIONS

We evaluate the models' ability to comply with explicit word-length constraints in image captioning tasks (see Table 2). Each model was prompted to generate a caption that did not exceed a specified word limit. The specific word limits we used were: 5, 11, 20, and 40 - which are used as $l_{uppper}$ values. Overall, compliance in this task was stronger than in the multiple-choice and binary question-answering settings. The only notable failure mode occurred under the strictest constraint—the 5-word limit—where most models struggled to adhere. Performance improved consistently as the allowed verbosity increased. Taken together with the results from the multiple-choice and binary tasks, these findings suggest a broader challenge: models tend to struggle with stricter brevity constraints, and this difficulty is often amplified by minor variations in input phrasing. We acknowledge that both task complexity and instruction-following may contribute to the difficulty of generating five-word captions. To isolate these effects, we added the strict instruction: "You are STRICTLY REQUIRED to answer within the word count as specified. You CANNOT exceed it." Compliance improved even for low word counts, quantitatively capturing how model behavior is influenced by prompt phrasing; not just task difficulty. This reinforces a key contribution: OLIIV surfaces meaningful instruction failures, not just capability limitations.

### 4.4 STRUCTURED OUTPUT

Our structured output analysis provides deeper insight into how VLMs handle response constraints, exposing systematic errors missed by accuracy-based evaluations. Although many outputs appear well-formed, they often include subtle but critical issues such as missing fields, incorrect formatting, or extraneous elements. Models also struggle to enforce constraints consistently across formats: some succeed in JSON but fail in XML or YAML, while others apply text wrapping in one format but not another. This suggests format-specific instruction-following failures shaped by learned internal generation patterns. To capture these issues, we define five key failure types: **Text wrapping** (unnecessary encodings), **Incorrect formatting** (invalid field names or data types), **Empty elements**

(placeholders with no function), **Duplicate elements** (repeated fields in arrays), and **Parsing failures** (structurally invalid outputs).

### 4.4.1 TEXT WRAPPING: A PERSISTENT STRUCTURAL BIAS

Prior work, notably IfEval Zhou et al. (2023), identified that models frequently introduce unnecessary text wrapping in structured outputs. Our results confirm that this remains a significant issue for VLMs as well, with analysis revealing how models apply wrapping inconsistently across formats. As shown in Table 3, parsing accuracy is substantially lower for raw (unprocessed) model outputs due to extraneous text that frequently wraps the structured content. This additional wrapping—such as introductory explanations or trailing punctuation—causes otherwise valid outputs to fail format-specific parsing. However, the behavior is not uniform across models or formats. For example, GPT-4o and Claude-3.7 Sonnet often wrap JSON outputs with extra text and exhibit similar issues in XML and YAML. In contrast, models like InternVL2.5-1B and Intern-3-1B display inconsistent wrapping behavior across structured formats. After removing this extraneous wrapping through lightweight post-processing, parsing accuracy improves dramatically. Models such as Gemini-2.0 Flash, Gemini-1.5 Flash, and Qwen2-VL-7B achieve near-perfect parsing rates across all formats. This suggests that these failures stem not from structural generation limitations but from learned formatting tendencies carried over from pretraining or alignment phases. While post-processing can recover performance in many cases, it raises concerns about the usability and reliability of raw model outputs in downstream pipelines. Furthermore, improvements are not universal: models like LLaMA-3.2-11B-VI and QwenVL still fail in key formats even after wrapping removal. This indicates that post-processing heuristics are often model- and format-specific, undermining their generalizability over time and across deployment settings.

Table 3: Parsing accuracy before and after wrapper removal. Scores range from 0.0 to 1.0. Raw outputs often fail due to extraneous text wrapping. Cleaning improves performance, highlighting the need for post-processing. 'Raw' columns show unprocessed outputs; 'Clean' columns show results after wrapper removal.

| Model | $JSON_{raw}$ | $XML_{raw}$ | $YAML_{raw}$ | $JSON_{clean}$ | $XML_{clean}$ | $YAML_{clean}$ |
|---|---|---|---|---|---|---|
| GPT-4o | 0.98 | 0.00 | 0.00 | 0.99 | 1.00 | 0.99 |
| GPT-4o Mini | 0.01 | 0.00 | 0.00 | 1.00 | 0.99 | 0.99 |
| Gemini-2.0 Flash | 0.00 | 0.00 | 0.00 | 1.00 | 0.99 | 1.00 |
| Gemini-2.0 Flash Lite | 0.00 | 0.00 | 0.00 | 1.00 | 0.99 | 1.00 |
| Gemini-1.5 Pro | 0.96 | 0.92 | 0.00 | 0.99 | 0.99 | 1.00 |
| Gemini-1.5 Flash | 0.97 | 0.89 | 0.00 | 1.00 | 1.00 | 1.00 |
| Claude-3-7 Sonnet | 0.93 | 0.01 | 0.00 | 1.00 | 0.95 | 0.99 |
| Claude-3.5 Sonnet | 1.00 | 1.00 | 0.99 | 1.00 | 0.99 | 0.99 |
| Intern-3-8B | 0.00 | 0.97 | 0.00 | 0.99 | 0.97 | 0.78 |
| Intern-3-1B | 0.59 | 0.03 | 0.35 | 0.95 | 0.03 | 0.35 |
| Intern-2.5-7B | 0.00 | 0.04 | 0.00 | 1.00 | 1.00 | 0.99 |
| Intern-2.5-1B | 0.97 | 0.96 | 0.87 | 0.97 | 0.96 | 0.89 |
| Llama-3.2-11B-VI | 0.74 | 0.27 | 0.23 | 0.74 | 0.27 | 0.23 |
| Llava-Next | 0.91 | 0.98 | 0.00 | 0.91 | 0.98 | 0.00 |
| Phi-4 | 0.99 | 0.99 | 0.00 | 0.99 | 0.99 | 0.005 |
| Phi-3.5 | 0.00 | 0.69 | 0.30 | 0.99 | 0.69 | 0.30 |
| Qwen2-VL-7B-Instruct | 0.00 | 1.00 | 0.93 | 1.00 | 1.00 | 0.93 |
| QwenVL | 0.00 | 0.00 | 0.99 | 0.00 | 0.00 | 0.99 |

### 4.4.2 STRUCTURED OUTPUT BREAKDOWN: REVEALING FORMAT-SPECIFIC FAILURES

Analyzing error types reveals fine-grained structural compliance issues (Table 4). GPT-4o inserts empty JSON elements in 74% of cases, reflecting over-reliance on templates. YAML is especially fragile: LLaVA-Next, Intern-3-8B, Intern-3-1B, InternVL2.5-1B, InternVL2.5-8B, Phi-4.0, and Phi-3.5 generate unparseable YAML in 100%, 22%, 65%, 100%, 37%, 11%, 99.5%, and 70% of cases, largely from spacing and nesting errors. LLaMA-3.2-11B prepends explanations to XML (73%) and YAML (77%), appends extraneous periods to JSON (26%), and omits XML headers (24%), an issue also seen in Intern-3-1B (97%) and Phi-3.5 (71%). Qwen-VL wraps JSON and XML outputs in arrays, breaking schema validity. LLaVA-Next, Gemini-1.5 Pro, and Gemini-1.5 Flash duplicate YAML elements in 14% and 5% of cases, while Intern-2.5-1B disrupts YAML structure by enumerating list items as keys (e.g., object 1: ...). These findings highlight the limits of parseability metrics and demonstrate OLIIV's ability to surface deeper compliance failures that impact usability.

Table 4: Structured output compliance is broken down by error type for models with significant formatting failures. Reported values are **success rates**, with lower scores (in red) indicating more frequent errors. For instance, GPT-4o leaves empty elements in 26% of JSON outputs, while LLaMA-3.2-11B-VI often prepends explanations or appends punctuation, reducing parse success across formats. Intern-3-1B and Phi-3.5 frequently omit XML headers, causing parse failures in 97% and 71% of cases. YAML is particularly error-prone: LLaVA-Next, Phi-4.0, and Intern variants commonly misalign indentation or misinterpret hierarchy.

| Model | Error Type | Json | Xml | Yaml |
|---|---|---|---|---|
| Phi-4.0 VI | Cannot be Parsed | - | - | 0.005 |
| Phi-3.5 VI | Cannot be Parsed | - | 0.69 | 0.30 |
| Intern-3-1B | Cannot be Parsed | - | 0.03 | 0.35 |
| Intern-2.5-7B | Cannot be Parsed | - | - | 0.63 |
| Intern-2.5-1B | Cannot be Parsed | - | - | 0.89 |
| Llama-3.2-11B-VI | Cannot be Parsed | 0.74 | 0.27 | 0.23 |
| Llava-Next | Cannot be Parsed | - | - | 0.0 |
| QwenVL | Cannot be Parsed | 0.00 | 0.00 | - |
| Intern-2.5-1B | Incorrect Formatting | - | - | 0.002 |
| Llama-3.2-11B-VI | Incorrect Formatting | - | - | 0.03 |
| GPT-4o | Empty Elements | 0.26 | - | - |
| gemini-1.5-flash | Element Duplications | - | - | 0.95 |
| gemini-1.5-pro | Element Duplications | - | - | 0.95 |
| Llava-Next | Element Duplications | 0.86 | - | - |

## 5 DISCUSSION

Our findings reveal a consistent pattern: superficial input variations often influence model compliance more than explicit instructions. This is most evident in the *Binary-Answer* and *Multiple-Choice* tasks, where models exhibit higher LIS scores—indicating greater verbosity—in the latter, even when explicitly instructed to select only the correct answer. In contrast, Binary-Answer prompts, which sometimes lack such constraints, result in more concise outputs. Structured output results reinforce this trend. For instance, GPT-4o inserts empty JSON elements in 74% of cases, indicating overfitting to learned structural templates. YAML formatting proves especially error-prone: InternVL2.5-7B, Phi-3.5, and Phi-4.0 fail to produce valid YAML in 37%, 70%, and 99.5% of cases, respectively, often due to indentation or nesting issues. Required XML declarations are missing in 97% of Intern-3-1B, 71% of Phi-3.5, and 24% of LLaMA-3.2-11B generations. The same LLaMA model appends extraneous punctuation to JSON outputs, breaking parseability in 26% of cases. QwenVL wraps structured outputs in arrays, violating expected schemas. Additionally, Intern-2.5-1B enumerates YAML list items as key-value pairs, rendering the output invalid.

These subtle but consistent errors reveal a broader issue: models often generate outputs that appear valid but violate instruction constraints, defaulting to pretrained formatting rather than following explicit directives. This undermines their reliability in settings requiring structural fidelity—such as benchmarking, pseudo-labeling, or automated decision support. Since models respond differently to superficially varied prompts, using a single prompt format may yield inconsistent evaluations. Systematic prompt variation is therefore essential for fair benchmarking. Thus, we introduce OLIIV.

While OLIIV reveals valuable insights into instruction-following behavior, it focuses on a fixed set of prompt variations and response formats. Future work should expand to additional task types, broader prompting conditions, and adversarial prompt strategies. Furthermore, the benchmark emphasizes text-based outputs; extending evaluation to more complex multimodal generation (e.g., diagrams, tables) remains an open challenge. Future work should explore adversarial prompt variation and stricter structural constraints to test generalization beyond format-specific memorization.

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

# 6 APPENDIX

## 6.1 POSITIONING OLIIV

Figure 3: OLIIV with respect to other instruction-following benchmarks.

| Benchmark | Output Format Adherence | Structured Output Constraints | Length Compliance | Prompt Variation Robustness | Evaluation Type | Modalities | Task Diversity |
|---|---|---|---|---|---|---|---|
| OLIIV | Yes – tests explicit answer formatting (e.g. A vs. I). | Yes – requires valid JSON, YAML, XML outputs. | Yes – enforces upper bounds on words (LIS metric). | Yes – prompt phrasing systematically varied. | Automatic (LIS, schema checks). | Vision + Language | Binary QA, multiple choice, structured output, and word-limited captioning. |
| MIA-Bench | Yes – layered instructions: include objects, follow style. | No – responses are free-text; structured format not required. | Yes – includes explicit length constraints. | No – no systematic rephrasing, but multiple simultaneous constraints. | LLM-based + human spot-checks. | Vision + Language | Captioning with layered constraints (style, content, length, grammar, tone). |
| VisIT-Bench | Partial – follows instruction intent, not strict formatting. | No – no structured format (JSON, etc.). | No – no word/character constraints enforced. | No – one prompt per query; no variants. | Reference-based + LLM alignment. | Vision + Language | 70+ task types from recognition to reasoning and generation. |
| MMMT-IF | Yes – global format constraints in multi-turn dialogues. | Partial – answer-style constraints, not JSON/XML. | Yes – many single-word/sentence instructions. | No – variation comes from dialog history, not prompt phrasing. | Automatic (PIF metric: code-based checks). | Vision + Language | Multi-turn Q&A with interleaved instruction constraints across ~20 turns. |
| IFEval | Yes – verifiable instructions (e.g. "400 words", use term). | Yes – some prompts require JSON or specific structure. | Yes – enforces length constraints explicitly. | No – no paraphrased variants; each prompt is atomic. | Fully automatic via code checks. | Text-Only | 500 prompts spanning 25 instruction types (length, content, style, format). |
| InFoBench (DRFR) | Yes – each instruction decomposed into binary checks. | Yes – supports formatting/structure checks. | Yes – length included among decomposed requirements. | No – no rewordings; each task has atomic multi-part instruction. | Mixed: DRFR metric + human/LLM validation. | Text-Only | 500 instructions → 2,250 sub-requirements across content, style, format, logic, and length. |

## 6.2 EXTENDED PROMPT VARIATION RESULTS

Table 5: Length Infidelity Scores (LIS) *Vars* results for multiple-choice and binary-answer tasks. These report the stricter variable-format compliance values you provided. Perfect scores of 0.0 are shaded green; higher scores are increasingly red.

| Model | MC (Vars) Letter | MC (Vars) Roman | Bin (Vars) Y/N | Bin (Vars) T/F |
|---|---|---|---|---|
| GPT-4o | 5.62 | 3.78 | 0.00 | 0.00 |
| GPT-4o Mini | 0.96 | 0.51 | 0.00 | 0.00 |
| Gemini-2.0 Flash Lite | 3.99 | 0.00 | 0.00 | 0.00 |
| Gemini-2.0 Flash | 0.47 | 0.06 | 0.00 | 0.00 |
| Gemini-1.5 Pro | 21.65 | 5.51 | 0.00 | 0.00 |
| Gemini-1.5 Flash | 12.25 | 0.16 | 0.00 | 0.00 |
| Claude-3-7 Sonnet | 13.81 | 9.42 | 0.00 | 0.00 |
| Claude-3.5 Sonnet | — | — | — | — |
| Intern-3-8B | 9.09 | 9.42 | 0.00 | 0.00 |
| Intern-3-1B | 6.86 | 6.53 | 0.00 | 0.00 |
| Intern-2.5-7B | 6.87 | 5.96 | 0.00 | 0.00 |
| Intern-2.5-1B | 6.93 | 7.66 | 0.00 | 0.00 |
| Llama-3.2-11B-VI | 11.04 | 10.15 | 0.00 | 0.00 |
| Llava-Next | 0.38 | 4.15 | 0.00 | 0.00 |
| Phi-4 | 1.22 | 3.05 | 0.00 | 0.00 |
| Phi-3.5 | 0.00 | 0.31 | 0.00 | 0.00 |
| Qwen2-VL-7B | — | — | — | — |
| QwenVL | 6.32 | 6.09 | 0.00 | 0.00 |

## 6.3 MULTIPLE CHOICE PROMPT VARIATIONS

"Select only the capital letter inside the parentheses corresponding to the most appropriate image caption for the given image from the following options: (A) A man leans forward and hits a ping pong ball. (B) Elephant with chair strapped to back and mouth open. (C) The batter swings his bat and hits the ball. (D) Two men with their heads touching, very intimate."

"From the following options: (A) A man leans forward and hits a ping pong ball. (B) Elephant with chair strapped to back and mouth open. (C) The batter swings his bat and hits the ball. (D) Two men with their heads touching, very intimate. Select only the capital letter inside the parentheses corresponding to the most appropriate image caption for the given image."

"Select only the Roman number inside the parentheses corresponding to the most appropriate image caption for the given image from the following options: (I) A man leans forward and hits a ping pong ball. (II) Elephant with chair strapped to back and mouth open. (III) The batter swings his bat and hits the ball. (IV) Two men with their heads touching, very intimate."

"From the following options:(I) A man leans forward and hits a ping pong ball. (II) Elephant with chair strapped to back and mouth open. (III) The batter swings his bat and hits the ball. (IV) Two men with their heads touching, very intimate. Select only the Roman number inside the parentheses corresponding to the most appropriate image caption for the given image."

### 6.3.1 ADDITIONAL VARIATIONS

"Choose the correct description by writing only its letter or Roman numeral."

"Your answer must be a single capital letter or Roman numeral. Which option correctly describes the image?"

"Only one caption matches the image. Answer with only its letter or Roman numeral."

### 6.3.2 STRICTNESS MODIFIER

"You are STRICTLY REQUIRED to answer with only one word as specified."

## 6.4 BINARY ANSWER PROMPT VARIATIONS

"Does the image show a fork? Please answer Yes or No only."

"Please only answer True or False according to the following statement about the image. The image shows a fork."

"Please answer True or False according to the following statement about the image. The image shows a fork."

### 6.4.1 ADDITIONAL VARIATIONS

"Your answer must be only either True or False. Consider the following statement about the image:"

"Decide if the following statement about the image is True or False, and respond with only one of those words:"

"Does this statement match the image? Reply with only True or False."

"Answer the question by writing only Yes or No."

"Is the answer Yes or No? Choose only one."

"Confirm the following: reply Yes if correct, No if not."

### 6.4.2 STRICTNESS MODIFIER

"You are STRICTLY REQUIRED to answer with only one word as specified."

## 6.5 STRUCTURED OUTPUT PROMPT VARIATIONS

"Write an image caption with no more than 1 sentence and a list of 2 objects present in the image. Your output must be only a parsable JSON string that the python function json.loads() can load successfully. Also, your output JSON string must follow the structure:

```
{"caption": "[the image caption]",
"objects": ["[object 1]", "[object 2]", "[object 3]", "[object 4]"]}''
```

"Write an image caption with no more than 1 sentence and a list of 2 objects present in the image. Your output must be a valid XML string that can be parsed successfully by the python function xml.etree.ElementTree.fromstring(). Your output XML string must have the following structure:

```
<?xml version="1.0"?>
<annotations>
    <caption>
        The image caption
    </caption>
<objects>
    <list>
        <item>object 1</item>
        <item>object 2</item>
        <item>object 3</item>
        <item>object 4</item>
    </list>
</objects>
</annotations>''
```

"Write an image caption with no more than 1 sentence and a list of 2 objects present in the image. Your output must be a string containing a valid YAML format that the python function yaml.safe_load() can successfully load the content. Your output string must have the following YAML structure:

```
caption:
    The image caption
```

```
objects:
    -object 1
    -object 2
    -object 3
    -object 4''
```

## 6.6 IMAGE CAPTION PROMPTS

"Write an image caption using no more than 5 words."

"Write an image caption using no more than 11 words."

"Write an image caption using no more than 20 words."

"Write an image caption using no more than 40 words."

### 6.6.1 STRICTNESS MODIFIER

"You are STRICTLY REQUIRED to answer within the word count as specified. You CANNOT exceed it."

## 6.7 PRIVATE MODEL SNAPSHOTS

Table 6: Model Versions and Snapshot Dates

| Model | Snapshot / Version |
|---|---|
| GPT-4o | 2024-08-06 |
| GPT-4o Mini | 2024-07-18 |
| Gemini-2.0 Flash Lite | 001 |
| Gemini-2.0 Flash | 001 |
| Gemini-1.5 Flash | 002 |
| Gemini-1.5 Pro | 002 |
| Claude-3-7 Sonnet | 2025-02-19 |
| Claude-3-5 Sonnet | 2024-10-22 |

## 6.8 RESPONSE ACCURACY

Table 7: Binary Choice Accuracy: Yes/No and True/False Scores

| Model | Yes/No ↑ | True/False ↑ |
|---|---|---|
| GPT-4o | 0.35 | 0.15 |
| GPT-4o Mini | 0.74 | 0.73 |
| Gemini-2.0 Flash Lite | 0.77 | 0.49 |
| Gemini-2.0 Flash | 0.67 | 0.27 |
| Gemini-1.5 Pro | 0.98 | 0.97 |
| Gemini-1.5 Flash | 0.67 | 0.28 |
| Claude-3-7 Sonnet | 0.61 | 0.40 |
| Claude-3.5 Sonnet | 0.97 | 0.97 |
| Intern-3-8B | 0.02 | 0.63 |
| Intern-3-1B | 0.15 | 0.69 |
| Intern | 0.98 | 0.98 |
| Llama-3.2-11B-Vision-Instruct | 0.97 | 0.97 |
| Llava-Next | 0.98 | 0.97 |
| Phi-4 | 0.97 | 0.97 |
| Phi-3.5 | 0.96 | 0.96 |
| Qwen2-VL-7B-Instruct | 0.98 | 0.98 |
| QwenVL | 0.98 | 0.98 |

Table 8: Multiple Choice Accuracy: Letter and Roman Number Scores

| Model | Letter ↑ | Roman Numeral ↑ |
|---|---|---|
| GPT-4o | 0.769 | 0.635 |
| GPT-4o Mini | 0.771 | 0.634 |
| Gemini-2.0 Flash Lite | 0.748 | 0.622 |
| Gemini-2.0 Flash | 0.749 | 0.623 |
| Gemini-1.5 Pro | 0.748 | 0.623 |
| Gemini-1.5 Flash | 0.748 | 0.623 |
| Claude-3-7 Sonnet | 0.595 | 0.504 |
| Claude-3.5 Sonnet | 0.761 | 0.599 |
| Intern-3-8B | 0.793 | 0.628 |
| Intern-3-1B | 0.762 | 0.644 |
| Intern-2.5-1B | 0.769 | 0.639 |
| Intern | 0.788 | 0.640 |
| Llama-3.2-11B-Vision-Instruct | 0.739 | 0.588 |
| Llava-Next | 0.774 | 0.634 |
| Phi-4 | 0.786 | 0.639 |
| Phi-3.5 | 0.771 | 0.610 |
| Qwen2-VL-7B-Instruct | 0.783 | 0.584 |
| QwenVL | 0.783 | 0.559 |

Table 9: BERT Precision, Recall, and F1 Scores for Image Captions

| Model | BERT Precision | BERT Recall | BERT F1 |
|---|---|---|---|
| gpt-4o | 0.8789 | 0.8734 | 0.8761 |
| gpt-4o-mini | 0.8810 | 0.8749 | 0.8779 |
| gemini-2.0-flash-lite | 0.8599 | 0.8519 | 0.8558 |
| gemini-2.0-flash | 0.8638 | 0.8532 | 0.8583 |
| gemini-1.5-pro | 0.8702 | 0.8535 | 0.8616 |
| gemini-1.5-flash | 0.8846 | 0.8654 | 0.8748 |
| claude-3-7-sonnet | 0.8738 | 0.8876 | 0.8805 |
| claude-3-5-sonnet | 0.8630 | 0.8787 | 0.8707 |
| intern-3-8b | 0.8932 | 0.8946 | 0.8938 |
| intern-3-1b | 0.8719 | 0.8924 | 0.8819 |
| intern-2.5-1b | 0.8929 | 0.8878 | 0.8902 |
| intern | 0.8856 | 0.8891 | 0.8873 |
| Llama-3.2-11B-Vision-Instruct | 0.9022 | 0.8957 | 0.8989 |
| llava-next | 0.8977 | 0.8859 | 0.8917 |
| phi4 | 0.9015 | 0.8947 | 0.8980 |
| phi3.5 | 0.8945 | 0.8865 | 0.8903 |
| Qwen2-VL-7B-Instruct | 0.8866 | 0.8952 | 0.8907 |
| qwenvl | 0.8556 | 0.8880 | 0.8714 |

Table 10: BERT Precision, Recall, and F1 Scores for Structured Output Captions

| Model | BERT Precision | BERT Recall | BERT F1 |
|---|---|---|---|
| GPT-4o | 0.7776 | 0.8844 | 0.8272 |
| GPT-4o Mini | 0.7688 | 0.8783 | 0.8192 |
| Gemini-2.0 Flash Lite | 0.7794 | 0.8783 | 0.8256 |
| Gemini-2.0 Flash | 0.7797 | 0.8785 | 0.8259 |
| Gemini-1.5 Pro | 0.7857 | 0.8837 | 0.8315 |
| Gemini-1.5 Flash | 0.7859 | 0.8831 | 0.8314 |
| Claude-3-7 Sonnet | 0.8065 | 0.8687 | 0.8363 |
| Claude-3.5 Sonnet | 0.7809 | 0.8888 | 0.8311 |
| Intern-3-8B | 0.7791 | 0.8834 | 0.8276 |
| Intern-3-1B | 0.7991 | 0.8876 | 0.8407 |
| Intern-2.5-1B | 0.7864 | 0.8703 | 0.8260 |
| Intern | 0.7785 | 0.8827 | 0.8269 |
| Llama-3.2-11B-Vision-Instruct | 0.8120 | 0.8857 | 0.8469 |
| Llava-Next | 0.7989 | 0.8812 | 0.8374 |
| Phi-4 | 0.7968 | 0.8761 | 0.8340 |
| Phi-3.5 | 0.7973 | 0.8841 | 0.8379 |
| Qwen2-VL-7B-Instruct | 0.7966 | 0.8857 | 0.8383 |
| QwenVL | 0.7710 | 0.8933 | 0.8272 |

