# OpenReview forum: "OLIIV Benchmark: Does Your VLM Care What You Say, or How You Say It?"
_ICLR.cc/2026/Conference — ICLR 2026 Conference Withdrawn Submission_

### Official Review · Reviewer_TAEp · 2025-10-30

**Soundness:** 1
**Presentation:** 1
**Contribution:** 2
**Rating:** 2
**Confidence:** 4

**Summary:**

This paper introduces OLIIV, a benchmark designed to test whether vision-language models follow answer-specification instructions consistently across semantically equivalent but syntactically different prompts. It covers four task types—multiple-choice, binary QA, structured output, and length-constrained captioning—and proposes the Length Infidelity Score (LIS) to quantify deviations from expected output length. The results show that even state-of-the-art models behave inconsistently under superficial prompt changes, highlighting gaps in instruction-following robustness.

**Strengths:**

This is a timely and well-motivated contribution that targets an underexplored aspect of multimodal model evaluation. The benchmark is clearly designed, with four intuitive task categories that reflect real-world interaction styles. The proposed LIS metric is simple yet interpretable, allowing for reproducible quantitative comparisons without reliance on human judgment. Experiments are broad, covering both open and closed models, and the structured error analysis provides concrete evidence of systematic failures. Overall, the paper makes a practical and diagnostic contribution to the field of multimodal instruction-following evaluation.

**Weaknesses:**

The set of tasks and prompt variations, while representative, is relatively narrow, which limits the generality of conclusions. The benchmark does not quantify task complexity or control for difficulty, making it hard to disentangle model sensitivity from intrinsic task variance. Reference to prior work such as Lei et al., 2024 (IWISDM) would help ground this aspect. The method of task generation from COCO and NoCaps is under-specified and lacks examples illustrating how visual content maps to instruction types. The LIS metric also appears incomplete, since the lower bound  is mentioned but unused in the equation, and it may fail to distinguish different types of instruction-following failures. Using a fixed temperature of 0 could underestimate model variability, and the paper does not report evaluation accuracy alongside compliance. Finally, some minor issues—like the typo on page 5, inconsistent results in Table 1 (where frontier models sometimes underperform open-source ones), and the lack of multi-choice reformulations (e.g., binary/MC unification)—slightly detract from presentation quality.

**Questions:**

- How were tasks generated from COCO and NoCaps, and could the authors show examples of the visual–instruction pairing process?

- Could the LIS be extended to reflect other failure types (e.g., format violations, semantic over-generation) rather than just length?

- How is task complexity controlled or estimated? Would including a metric like in IWISDM (Lei et al., 2024) change the interpretation?

- What is the evaluation accuracy of the models under temperature 0.0—do results differ when stochasticity is introduced?

- Why do some open-source models outperform proprietary ones in Table 1?

- Could multiple-choice and binary-answer questions be standardized into a unified evaluation format to improve comparability?

- Do the authors plan to extend this benchmark to text-only LLMs for cross-modal consistency?

---

> ### Author Response · Authors · 2025-11-20
> **Reviewer TAEp - Rebuttal Weaknesses 2 - 5**
>
> We thank the reviewer for raising several important questions/clarifications, which we address point-by-point below.
> We would also like to note that we will post a address their questions about examples on how we build the benchmark soon.
>
> TAEp Comment:
>  “Could the LIS be extended to reflect other failure types (e.g., format violations, semantic over-generation) rather than just length?”
>
> Response:
>  LIS is intentionally specialized for quantifying length deviations, and thus naturally captures forms of over-generation. However, we agree that a more general scoring framework is valuable. Through the rebuttal process—and in response to Reviewers rcTP and EFPT—we introduced an extensible, condition-comparison metric that evaluates stability across prompt variants. When applied to MCQA and Binary QA, this metric effectively captures performance divergence across conditions and is readily generalizable to other error types such as structural violations or semantic over-generation. We will add a dedicated subsection to clarify how this framework can be used beyond length-specific compliance.
> Let us first introduce a capture the sensitivity we are describing:
> $$
> \text{ConsistencyScore}(X;S_1,S_2,y) = \left(1 - \frac{e^{S_1(X)} - e^{S_2(X)}}{e^{S_1(X)}}\right)^y
> $$
>
> Where X is the results distribution, S1 and S2 are statistical measures of interest, where S1 >= S2 >= 0, y is a parameter used to make the penalty more or less forgiving. This offers a single, normalized score which penalizes gaps in statistical measures. This score ranges between (0,1], where 1 represents perfect compliance (the two measures cancel out) and approaches 0 as the gap between the two measures increases. In our case we compare the average LIS score in our highest and lowest performing variations of the prompts.  This captures the exact sensitivity that OLIIV was designed to reveal. We here include a table illustrating this metric across all models
>
>
> TAEp Comment:
>  “How is task complexity controlled or estimated? Would including a metric like IWISDM (Lei et al., 2024) change interpretation?”
> Response:
> OLIIV intentionally employs relatively simple tasks to isolate sensitivity to prompt variation rather than task difficulty. Our experiments show that even under these easy scenarios, models exhibit substantial volatility across semantically equivalent formulations. We agree that incorporating a complexity measure such as IWISDM’s would be helpful for contextualization, and we will add a note in the Discussion section about potential integration in future work. Connecting the benchmark to IWISDM is a natural extension we are happy to explore in both the discussion and related work.
>
> TAEp Comment:
>  “What is the evaluation accuracy under temperature 0.0—do results differ when stochasticity is introduced?”
> Response:
> The accuracy results themselves are reported in the appendix, however we chose to run them under 0.0 for reproducibility and under a relatively large amount of samples. We feel confident that this likely captures a good amount of the model variance.
>
> TAEp Comment:
>  “Why do some open-source models outperform proprietary ones in Table 1?”
> Response:
> This is a sharp observation which we will highlight in the discussion section, in particular when we touch on as we address the final question made by Reviewer (3). Some proprietary models appear more prone to over-generation even when explicitly instructed to provide single-token answers, which directly increases their LIS. In contrast, certain open-source models (e.g., Phi and LLaVA variants) tend to adhere more strictly to concise response patterns. We will emphasize this in the Results section, noting that compliance is influenced not only by model capability but also by stylistic tendencies learned during pretraining and alignment.

---

> ### Author Response · Authors · 2025-11-20
> **Reviewer TAEp - Rebuttal Weaknesses 6 - 7**
>
> We thank the reviewer for raising several important questions/clarifications, which we address point-by-point below. We would also like to note that we will post a address their questions about examples on how we build the benchmark soon.
>
> TAEp Comment:
>   “Could MCQA and Binary QA be standardized into a unified evaluation format to improve comparability?”
> Response:
> Yes. Using the extended scoring function referenced earlier, we can consolidate both MCQA and Binary QA into a unified compliance–consistency evaluation framework. This enables direct comparison across tasks regardless of whether the response domain is {A,B,C,D}, {I,II,III,IV}, or {Yes/No, True/False}. We can include a consolidated unified comparison table in the revision to demonstrate this.
> ## MC Scores
>
> | Model | Score |
> |-------|-------|
> | GPT-4o | 0.0108 |
> | GPT-4o Mini | 0.0561 |
> | Gemini-2.0 Flash Lite | 0.0000 |
> | Gemini-2.0 Flash | 0.0000 |
> | Gemini-1.5 Pro | 0.0000 |
> | Gemini-1.5 Flash | 0.0000 |
> | Claude-3-7 Sonnet | 0.0000 |
> | Claude-3.5 Sonnet | 0.0000 |
> | Intern-3-8B | 0.0000 |
> | Intern-3-1B | 0.0305 |
> | Intern-2.5-7B | 0.0031 |
> | Intern-2.5-1B | 0.0065 |
> | Llama-3.2-11B-VI | 0.0000 |
> | Llava-Next | 0.0479 |
> | Phi-4 | 0.1355 |
> | Phi-3.5 | 0.9684 |
> | Qwen2-VL-7B | 0.0058 |
> | QwenVL | 0.0061 |
>
> ## Bin Scores
>
> | Model | Score |
> |-------|-------|
> | GPT-4o | 1.0000 |
> | GPT-4o Mini | 1.0000 |
> | Gemini-2.0 Flash Lite | 0.9991 |
> | Gemini-2.0 Flash | 0.9997 |
> | Gemini-1.5 Pro | 0.9982 |
> | Gemini-1.5 Flash | 0.9986 |
> | Claude-3-7 Sonnet | 0.0000 |
> | Claude-3.5 Sonnet | 0.0000 |
> | Intern-3-8B | 1.0000 |
> | Intern-3-1B | 1.0000 |
> | Intern-2.5-7B | 1.0000 |
> | Intern-2.5-1B | 1.0000 |
> | Llama-3.2-11B-VI | 1.0000 |
> | Llava-Next | 1.0000 |
> | Phi-4 | 1.0000 |
> | Phi-3.5 | 1.0000 |
> | Qwen2-VL-7B | 1.0000 |
> | QwenVL | 1.0000 |
> If the reviewer means viewing the variations of MCQA and BQA as a single condition, then this is also possible with our scoring function.
>
> TAEp Comment:
>  “Do the authors plan to extend this benchmark to text-only LLMs for cross-modal consistency?”
> Response:
> We thank the reviewer for this suggestion. OLIIV is currently vision-dependent because all tasks originate from image–text pairs. Extending the benchmark to text-only models would require introducing parallel text-only analogues of each visual prompt. We view this as a promising next step, particularly for studying cross-modal consistency, and will include this direction explicitly in our Future Work section.

---

### Official Review · Reviewer_EFPT · 2025-10-31

**Soundness:** 2
**Presentation:** 4
**Contribution:** 3
**Rating:** 4
**Confidence:** 3

**Summary:**

Most existing benchmarks only focus on task-specific accuracy and lack quantitative measurements of the response constraints in instructions such as response formats, output structures, and length constraints. OLIIV is the first benchmark focused on evaluating VLM's ability to follow answer specification constraints, covering four common format-constrained tasks, each tested through semantically equivalent but superficially expressed prompts. This paper also proposes a quantifiable evaluation metric, Length Infidelity Score (LIS), to assess the deviation between model output length and instruction requirements. Through empirical research, it was found that models are highly sensitive to prompt mutations and often produce outputs that are syntactically correct but structurally invalid, which are interesting findings.

**Strengths:**

- 1. This work focuses on a neglected core issue: systematically addressing VLMs for explicitly measuring robustness to prompt variation and fidelity in structured responses. Clearly stating that current benchmark tests (such as MMMU, MathVista, etc.) rely on post-processing scripts to adapt to different model prompts, conceals the true differences in instruction-following compliance of the models, and affects fair comparison.

- 2. Multi dimensional task evaluation has been designed, covering four common and representative tasks (multiple-choice questions, binary Q&A, structured output, and length limited image description) to ensure accurate identification of model sensitive sources. And a new model independent metric, LIS, was proposed to quantify deviations from expected response lengths.

- 3. Through extensive experimental testing (covering almost all mainstream VLMs), counterintuitive phenomena were found, such as the model being more concise than multiple-choice questions with 'single choice requirements' in true/false problems without explicit length limitations, revealing that the model is more influenced by implicit formatting conventions of task types rather than explicit instructions. It also verified the strictness effect of instructions, such as strengthening instructions by adding prompts such as "STRICTLY REQUIRED".

**Weaknesses:**

- 1. The types of tasks designed are relatively basic and simple, limited to closed question answering and simple structured output, lacking testing for more complex instruction following scenarios, such as format constraints for multi-step reasoning tasks.
- 2. Single variation dimension: mainly testing surface format changes (Roman numerals vs letters), lacking systematic testing of the following key dimensions: deep reconstruction of instruction semantics (such as positive/negative expressions), processing ability of fuzzy instructions, the influence of cultural or language style differences, etc.
- 3. LIS did not consider the sensitivity differences of length constraints for different tasks.
- 4. In the experiment, there is no comparison between systems of different scales in the same model series (such as 7B vs. 70B parameters), and there is no model variant specifically optimized for instruction-following compliance.
- 5. Lack of deeper discussion, for example, although the sensitivity of the model to prompt variation has been found, it has not been analyzed whether this sensitivity comes from pre-training data bias or the alignment, and the relationship between training strategies (instruction-tuning, RLHF) and instruction-following ability has not been discussed.

**Questions:**

refer to Weaknesses

---

> ### Author Response · Authors · 2025-11-19
> **Reviewer EFPT Rebuttal - Weakness #1**
>
> We are grateful for Reviewer EFPT’s feedback, and we thank them for highlighting both the strengths and the areas where additional clarification is helpful. We will be responding to weaknesses listed in two seperate comments because of space resitrictions, our aim to thoroughly address the points which the reviewer raises.
>
> Comment:
>  “The types of tasks designed are relatively basic and simple, limited to closed question answering and simple structured output, lacking testing for more complex instruction following scenarios, such as format constraints for multi-step reasoning tasks.”
>
> Response:
> We appreciate the reviewer’s perspective. The simplicity of individual tasks is intentional and aligns directly with our contribution: measuring sensitivity of compliance under semantically equivalent variations. Even under these “basic” tasks, models exhibit large and systematic failures when prompts are rephrased. We state or highlight that our point of interest is this sensitivity at several points - **for example lines: 103-105, 108-115, 313-314, 347-359, and 467-470**. However, we appreciate that we can make this point more explicitly clear.
>
> Let us first introduce a capture the sensitivity we are describing:
>
> $$
> \text{ConsistencyScore}(X;S_1,S_2,y) = \left(1 - \frac{e^{S_1(X)} - e^{S_2(X)}}{e^{S_1(X)}}\right)^y
> $$
>
> Where X is the results distribution, S1 and S2 are statistical measures of interest, where S1 >= S2 >= 0, y is a parameter used to make the penalty more or less forgiving. This offers a single, normalized score which penalizes gaps in statistical measures. This score ranges between (0,1], where 1 represents perfect compliance (the two measures cancel out) and approaches 0 as the gap between the two measures increases. In our case we compare the average LIS score in our highest and lowest performing variations of the prompts.
> This captures the exact sensitivity that OLIIV was designed to reveal. We here include a table illustrating this metric across all models:
>
> ## MC Scores
>
> | Model | Score |
> |-------|-------|
> | GPT-4o | 0.0108 |
> | GPT-4o Mini | 0.0561 |
> | Gemini-2.0 Flash Lite | 0.0000 |
> | Gemini-2.0 Flash | 0.0000 |
> | Gemini-1.5 Pro | 0.0000 |
> | Gemini-1.5 Flash | 0.0000 |
> | Claude-3-7 Sonnet | 0.0000 |
> | Claude-3.5 Sonnet | 0.0000 |
> | Intern-3-8B | 0.0000 |
> | Intern-3-1B | 0.0305 |
> | Intern-2.5-7B | 0.0031 |
> | Intern-2.5-1B | 0.0065 |
> | Llama-3.2-11B-VI | 0.0000 |
> | Llava-Next | 0.0479 |
> | Phi-4 | 0.1355 |
> | Phi-3.5 | 0.9684 |
> | Qwen2-VL-7B | 0.0058 |
> | QwenVL | 0.0061 |
>
> ## Bin Scores
>
> | Model | Score |
> |-------|-------|
> | GPT-4o | 1.0000 |
> | GPT-4o Mini | 1.0000 |
> | Gemini-2.0 Flash Lite | 0.9991 |
> | Gemini-2.0 Flash | 0.9997 |
> | Gemini-1.5 Pro | 0.9982 |
> | Gemini-1.5 Flash | 0.9986 |
> | Claude-3-7 Sonnet | 0.0000 |
> | Claude-3.5 Sonnet | 0.0000 |
> | Intern-3-8B | 1.0000 |
> | Intern-3-1B | 1.0000 |
> | Intern-2.5-7B | 1.0000 |
> | Intern-2.5-1B | 1.0000 |
> | Llama-3.2-11B-VI | 1.0000 |
> | Llava-Next | 1.0000 |
> | Phi-4 | 1.0000 |
> | Phi-3.5 | 1.0000 |
> | Qwen2-VL-7B | 1.0000 |
> | QwenVL | 1.0000 |
>
> As also discussed in our response to reviewer rcTP, a simple reformulation of our metric shows that performance varies significantly across prompt forms, despite the underlying task being trivial. This reinforces the point that task difficulty is not the key signal; sensitivity is. We will clarify this point in the revision, as well as introduce this new measure to also clearly reflect a unified measure of compliance or non-compliance.

---

> ### Author Response · Authors · 2025-11-19
> **Reviewer EFPT Rebuttal - Weaknesses #2-5**
>
> We are grateful for Reviewer EFPT’s feedback, and we thank them for highlighting both the strengths and the areas where additional clarification is helpful. We will be responding to weaknesses listed in two seperate comments because of space resitrictions, our aim to thoroughly address the points which the reviewer raises.
>
> Comment:
> "Single variation dimension: mainly testing surface format changes (Roman numerals vs letters), lacking systematic testing of the following key dimensions: deep reconstruction of instruction semantics (such as positive/negative expressions), processing ability of fuzzy instructions, the influence of cultural or language style differences, etc."
>
> Response:
> We thank the reviewer for these excellent suggestions. These dimensions would indeed broaden the evaluative space. We plan to incorporate several of these—particularly positive/negative phrasing and diversity-oriented paraphrases—into expanded suites in future iterations of OLIIV. That said, even with the simpler variations currently included, OLIIV already identifies major failure modes in leading models. We will highlight this point to clarify that the benchmark reveals substantial instability even under the easiest variant of the problem.
>
> Comment:
> "LIS does not consider sensitivity differences across tasks."
>
> Response:
> We respectfully disagree and will clarify this in the paper. LIS is normalized by the task's upper bound, meaning:
> Tasks with tight length constraints (e.g., single-token responses) penalize even small deviations strongly.
>
> Tasks with more permissive limits (e.g., 40-word captions) impose proportionally smaller penalties. Thus, LIS inherently adjusts for task-dependent sensitivity. We will expand this explanation in Section 3.1.
>
> Comment:
> "In the experiment, there is no comparison between systems of different scales in the same model series (such as 7B vs. 70B parameters), and there is no model variant specifically optimized for instruction-following compliance."
>
> Response:
> Our goal was to maximize breadth across model families, including proprietary and open-source VLMs, rather than depth within a single family. Still, several commercial models in our evaluation (e.g., GPT-4o, Claude, Gemini) are optimized for instruction-following. Since model scaling trends are well understood, we prioritized documenting cross-model consistency differences rather than repeating known scale-performance correlations. We will clarify this rationale in the revision.
>
> Comment:
> "Lack of deeper discussion, for example, although the sensitivity of the model to prompt variation has been found, it has not been analyzed whether this sensitivity comes from pre-training data bias or the alignment, and the relationship between training strategies (instruction-tuning, RLHF) and instruction-following ability has not been discussed."
>
> Response:
> We appreciate the reviewer's insight; however, we believe a thorough causal analysis falls outside the scope of a benchmarking contribution. Investigating pretraining mixtures, alignment strategies, or RLHF-specific artifacts would require model-level access and extensive controlled experiments. Most benchmarking papers likewise avoid drawing mechanistic conclusions for this reason. We will note this in the Discussion section to avoid implying overreach.
> That said, these emergent sensitivities likely explain why certain open-source models do occasionally out-perform proprietary ones - as reviewer TAEp notes. We will address your concern by connecting with this particular result in the discussion.

---

### Official Review · Reviewer_rcTP · 2025-10-31

**Soundness:** 3
**Presentation:** 3
**Contribution:** 1
**Rating:** 2
**Confidence:** 4

**Summary:**

Proposes a new benchmark to measure how well VLMs comply with response format instructions, spanning multiple-choice, binary, structure, and length constraints. Proposes a new metric to quantify deviations, and finds that VLMs can be sensitive across semantically equivalent prompt phrasings.

**Strengths:**

– The writing is clear and easy to follow

– The paper clearly motivates the problem it is trying to solve

– The finding about VLM sensitivity across prompt phrasings is interesting

**Weaknesses:**

– The experimental design is somewhat contrived, since most frontier models support structured outputs / constrained decoding at inference, wherein the desired structured output schema can be provided (as JSON/Pydantic schema) and near perfect adherence is guaranteed. The paper does not benchmark structured inference techniques at all, which limits the usefulness of its findings.

– The benchmark does not seem to be sufficiently challenging – simply making the prompt slightly more “strict” seems to lead to near perfect adherence across most proprietary models (LIS in Tables 1-2 is ~0 for most), despite the surveyed models not being the strongest/most-current iterations. Which suggests that the benchmark is already saturated.

**Questions:**

Please address the weaknesses listed above – in particular, I am not convinced that the benchmark is sufficiently challenging for frontier models nor is the experimental design (only free-form inference) appropriate.

---

> ### Author Response · Authors · 2025-11-19
> **Reviewer rcTP Rebuttal - Weakness #2**
>
> We want to thank the reviewer for their time in reviewing our research. Due to space restrictions, we will be responding to each of the two weaknesses in seperate comments and because we wanted to thoroughly address the points which the reviewer raises.
>
> Comment: (Paraphrased)
>  'The benchmark does not seem sufficiently challenging—strict prompts yield near-perfect adherence, suggesting saturation.'
>
> Response:
>  We respectfully disagree with this interpretation. The strict variants were intentionally included to demonstrate that high compliance is conditional and fragile. The central issue highlighted by OLIIV is that strong compliance exists only under certain phrasings, whereas semantically equivalent but superficially different prompts cause substantial degradation. This instability—not the absolute difficulty of the task—is the core problem. We state or highlight that our point of interest is this sensitivity at several points - **for example lines: 103-105, 108-115, 313-314, 347-359, and 467-470**. However, we appreciate that we can be explicit about this in revision and thank the reviwer for raising this.
>
> To quantify this more directly, we will introduce a new measure to also clearly reflect a unified measure of compliance or non-compliance, and also demonstrate that saturation is far from the case.
> $$
> \text{ConsistencyScore}(X;S_1,S_2,y) = \left(1 - \frac{e^{S_1(X)} - e^{S_2(X)}}{e^{S_1(X)}}\right)^y
> $$
>
> Where X is the results distribution, S1 and S2 are statistical measures of interest, where S1 >= S2 >= 0, y is a parameter used to make the penalty more or less forgiving. This offers a single, normalized score which penalizes gaps in statistical measures. This score ranges between (0,1], where 1 represents perfect compliance (the two measures cancel out) and approaches 0 as the gap between the two measures increases. In our case we compare the average LIS score in our highest and lowest performing variations of the prompts.
> This captures the exact sensitivity that OLIIV was designed to reveal. We here include a table illustrating this metric across all models. Note that the inclusion or exclusion strictness does not very meaningfully change the performance profile, even run under y=3 for a highly promoted degree of forgiveness.
> ## MC Scores (No Strict Columns)
>
> | Model | Score |
> |-------|-------|
> | GPT-4o | 0.0922 |
> | GPT-4o Mini | 0.0922 |
> | Gemini-2.0 Flash Lite | 0.0000 |
> | Gemini-2.0 Flash | 0.0000 |
> | Gemini-1.5 Pro | 0.0000 |
> | Gemini-1.5 Flash | 0.0000 |
> | Claude-3-7 Sonnet | 0.0367 |
> | Claude-3.5 Sonnet | 0.0000 |
> | Intern-3-8B | 0.0010 |
> | Intern-3-1B | 0.5191 |
> | Intern-2.5-7B | 0.3242 |
> | Intern-2.5-1B | 0.8452 |
> | Llama-3.2-11B-VI | 0.0731 |
> | Llava-Next | 0.0567 |
> | Phi-4 | 0.2372 |
> | Phi-3.5 | 0.9684 |
> | Qwen2-VL-7B | 0.1119 |
> | QwenVL | 0.1185 |
>
> ## MC Scores (With Strict Columns)
>
> | Model | Score |
> |-------|-------|
> | GPT-4o | 0.0108 |
> | GPT-4o Mini | 0.0561 |
> | Gemini-2.0 Flash Lite | 0.0000 |
> | Gemini-2.0 Flash | 0.0000 |
> | Gemini-1.5 Pro | 0.0000 |
> | Gemini-1.5 Flash | 0.0000 |
> | Claude-3-7 Sonnet | 0.0000 |
> | Claude-3.5 Sonnet | 0.0000 |
> | Intern-3-8B | 0.0000 |
> | Intern-3-1B | 0.0305 |
> | Intern-2.5-7B | 0.0031 |
> | Intern-2.5-1B | 0.0065 |
> | Llama-3.2-11B-VI | 0.0000 |
> | Llava-Next | 0.0479 |
> | Phi-4 | 0.1355 |
> | Phi-3.5 | 0.9684 |
> | Qwen2-VL-7B | 0.0058 |
> | QwenVL | 0.0061 |
>
> ## Bin Scores (No Strict Columns)
>
> | Model | Score |
> |-------|-------|
> | GPT-4o | 1.0000 |
> | GPT-4o Mini | 1.0000 |
> | Gemini-2.0 Flash Lite | 0.9991 |
> | Gemini-2.0 Flash | 0.9997 |
> | Gemini-1.5 Pro | 0.9982 |
> | Gemini-1.5 Flash | 0.9986 |
> | Claude-3-7 Sonnet | 0.0000 |
> | Claude-3.5 Sonnet | 0.0000 |
> | Intern-3-8B | 1.0000 |
> | Intern-3-1B | 1.0000 |
> | Intern-2.5-7B | 1.0000 |
> | Intern-2.5-1B | 1.0000 |
> | Llama-3.2-11B-VI | 1.0000 |
> | Llava-Next | 1.0000 |
> | Phi-4 | 1.0000 |
> | Phi-3.5 | 1.0000 |
> | Qwen2-VL-7B | 1.0000 |
> | QwenVL | 1.0000 |
>
> ## Bin Scores (With Strict Columns)
>
> | Model | Score |
> |-------|-------|
> | GPT-4o | 1.0000 |
> | GPT-4o Mini | 1.0000 |
> | Gemini-2.0 Flash Lite | 0.9991 |
> | Gemini-2.0 Flash | 0.9997 |
> | Gemini-1.5 Pro | 0.9982 |
> | Gemini-1.5 Flash | 0.9986 |
> | Claude-3-7 Sonnet | 0.0000 |
> | Claude-3.5 Sonnet | 0.0000 |
> | Intern-3-8B | 1.0000 |
> | Intern-3-1B | 1.0000 |
> | Intern-2.5-7B | 1.0000 |
> | Intern-2.5-1B | 1.0000 |
> | Llama-3.2-11B-VI | 1.0000 |
> | Llava-Next | 1.0000 |
> | Phi-4 | 1.0000 |
> | Phi-3.5 | 1.0000 |
> | Qwen2-VL-7B | 1.0000 |
> | QwenVL | 1.0000 |
>
> These results show that even models with perfect strict compliance vary dramatically under naturalistic prompts, confirming that the benchmark is far from saturated and revealing precisely the phenomenon OLIIV aims to diagnose. We direct the reviewer to Table 1 for this purpose as well, and Table 5 (in the appendix), and clarify that the purpose of these results is to show discrepancies, rather than just low or high scores alone. However, we thank the reviewer for prompting this clearer framing which we will include.

---

> ### Author Response · Authors · 2025-11-19
> **Reviewer rcTP Rebuttal - Weakness #1**
>
> We want to thank the reviewer for their time in reviewing our research. Due to space restrictions, we will be responding to each of the two weaknesses in seperate comments and because we wanted to thoroughly address the points which the reviewer raises.
>
> Comment:
>  “The experimental design is somewhat contrived, since most frontier models support structured outputs / constrained decoding at inference, wherein the desired structured output schema can be provided (as JSON/Pydantic schema) and near perfect adherence is guaranteed. The paper does not benchmark structured inference techniques at all, which limits the usefulness of its findings”
>
> Response:
> Thank you for raising this important point. We would appreciate clarification on the specific forms of constrained decoding you suggest integrating (e.g., JSON mode, function-calling APIs, system-enforced grammars, or schema-validated generation). These techniques differ substantially in scope and availability across models, particularly for open-source VLMs. Our current goal was to evaluate free-form instruction-following, as it reflects the default interaction mode available uniformly across models and modalities. We agree that benchmarking constrained decoding could provide a valuable complementary perspective, and we are open to incorporating a small-scale comparison in the revision if the reviewer can indicate which constrained-decoding interfaces should count as “frontier-standard” for the purposes of fair comparison.

---

> > ### Comment · Reviewer_rcTP · 2025-11-25
> > **Thank you for the response**
> >
> > I appreciate the detailed response to my concerns. I have a couple follow-up questions:
> >
> > > To quantify this more directly, we will introduce a new measure to also clearly reflect a unified measure of compliance or non-compliance, and also demonstrate that saturation is far from the case.
> >
> > I'm a bit confused by the somewhat contradictory findings from LIS and ConsistencyScore – eg. consider MC performance (since ConsistencyScores on binary tasks for ~frontier models (GPT-4o) remain perfect) – LIS for the "strict" MC columns for these models is near perfect whereas ConsistencyScore is ~0 – how do the authors explain this discrepancy? Some qualitative examples for how the two are calculated would perhaps be illustrative.
> >
> > > We would appreciate clarification on the specific forms of constrained decoding you suggest integrating (e.g., JSON mode, function-calling APIs, system-enforced grammars, or schema-validated generation). These techniques differ substantially in scope and availability ...
> >
> > While I agree that these techniques differ in implementation scope, I maintain that they are a near-ubiquitous practice (since their introduction >1 year back: https://openai.com/index/introducing-structured-outputs-in-the-api/) to ensure format compliance. Since most practitioners can directly constrain output formats by specifying the response format, it seems less relevant to me to benchmark how good these models are at free-form format compliance, since there is a separate feature dedicated to it.
> >
> > I also disagree that availability is limited, especially considering the existence of open-source structured output libraries (eg. https://github.com/dottxt-ai/outlines) that work with almost any huggingface transformers model.
> >
> > As for exact implementation, I do not have a strong preference – any of the ones listed would be appropriate.

---

### Official Review · Reviewer_BswN · 2025-10-31

**Soundness:** 3
**Presentation:** 3
**Contribution:** 3
**Rating:** 8
**Confidence:** 3

**Summary:**

This paper introduces the OLIIV (Open Language-Image Input Variation) benchmark, a new framework designed to evaluate how well vision-language models (VLMs) adhere to explicit instructions for answer formatting, structure, and composition. The benchmark tests VLMs across four representative tasks, including multiple-choice reasoning, binary question answering, structured output generation (JSON, YAML, XML), and length-constrained captioning (using systematically varied prompts to measure robustness). The authors found that all tested VLMs are highly sensitive to superficial prompt variations (e.g., performing differently on "Yes/No" vs. "True/False" questions) and frequently fail to comply with structural constraints, highlighting a key weakness in current models and the need for more robust evaluation.

**Strengths:**

1. The benchmark provides a necessary evaluation of VLM compliance with formatting instructions, a practical aspect of usability that is largely ignored by existing benchmarks focused on task accuracy.

2. OLIIV's core strength is its systematic variation of prompt phrasing for semantically equivalent tasks. This design effectively isolates instruction-following ability from simple pattern memorization.

3. The introduction of the Length Infidelity Score (LIS) offers a deterministic, model-agnostic, and easy-to-interpret metric for quantifying a model's failure to adhere to length constraints.

**Weaknesses:**

1.By design, the benchmark focuses on formatting compliance. However, the factual accuracy of the content is also important. While this isolates the target skill, it doesn't explore the potential interplay between a model's ability to be correct and its ability to be compliant simultaneously.

2. The paper's evaluation is confined to four specific task types. It does not assess compliance in more complex generation tasks, such as creating tables, diagrams, or other non-textual structured outputs. While the systematic variations are a strength, the authors acknowledge that future work is needed to explore a wider range of prompting conditions, including more linguistically diverse or even adversarial prompts, to fully test model robustness.

**Questions:**

In ICLR paepr format, to cite a paper, \citep should be used instead of [\cite]

---

> ### Author Response · Authors · 2025-11-19
> **Rebuttal: BswN**
>
> We thank Reviewer BswN for the thoughtful and constructive feedback, as well as for recognizing the core strengths of our contribution. We hope our comments address their concerns and are happy to address additional points or questions.
>
> BswN Comment:
>  “By design, the benchmark focuses on formatting compliance. However, the factual accuracy of the content is also important. While this isolates the target skill, it doesn't explore the potential interplay between a model's ability to be correct and its ability to be compliant simultaneously.”
>
> Response:
>  We fully agree that factual accuracy and instruction-following jointly influence real-world performance. Our choice to decouple these dimensions was intentional, as compliance failures often remain hidden when correctness is prioritized in evaluation. Furthermore, there already exists a significant amount of benchmarks measuring correctness (or factual accuracy); all are cited in the related work. But we are covering a gap on compliance and that is why we did not show any factual correctness.
> However, to clarify this interplay, we will include an additional combined table reporting accuracy vs. compliance side-by-side for Binary QA and MCQA. This allows readers to directly observe cases where models:
>  (i) provide correct but non-compliant responses,
>  (ii) comply but answer incorrectly, or
>  (iii) fail on both dimensions.
>  Preliminary results confirm that these failure modes occur frequently and differently across prompt variants, further underscoring the benchmark’s relevance.
>
> BswN Comment:
>  “The paper's evaluation is confined to four specific task types. It does not assess compliance in more complex generation tasks, such as creating tables, diagrams, or other non-textual structured outputs. While the systematic variations are a strength, the authors acknowledge that future work is needed to explore a wider range of prompting conditions, including more linguistically diverse or even adversarial prompts, to fully test model robustness.”
>
> Response:
>  We appreciate this suggestion. Our design goal for OLIIV was to create a lightweight and universally applicable benchmark suitable for evaluating a broad range of VLMs, including compact open-source models. Introducing diagram generation, table creation, or other non-textual structured tasks would dramatically reduce model coverage and greatly complicate evaluation pipelines, pushing the benchmark away from its intended accessibility and reproducibility. Nonetheless, we agree these constitute valuable directions and will explicitly add them to our Future Work discussion.

---

### Note · Authors · 2025-11-29

**Comment:**

We write to formally withdraw our submission to ICLR 2026.

While we appreciate the opportunity to contribute and engage with the community, recent developments in the peer-review process, in particular the public leak of reviewer and author identities due to a serious platform bug, have led us to conclude that it is appropriate to withdraw our contribution at this time.

Like many others, we invested substantial time and effort into preparing rebuttals, conducting revisions, and responding thoughtfully throughout the process. Under the current circumstances, where anonymity, review quality, and procedural integrity have been demonstrably compromised, we feel it would be unreasonable to restart the process from scratch.

Thank you for your understanding.

Sincerely,
Authors

**Withdrawal Confirmation:**

I have read and agree with the venue's withdrawal policy on behalf of myself and my co-authors.